# High Serum Levels of CCL20 Are Associated with Recurrence and Unfavorable Overall Survival in Advanced Melanoma Patients Receiving Immunotherapy

**DOI:** 10.3390/cancers16091737

**Published:** 2024-04-29

**Authors:** Julian Kött, Inka Lilott Hoehne, Isabel Heidrich, Noah Zimmermann, Kim-Lea Reese, Tim Zell, Glenn Geidel, Alessandra Rünger, Stefan W. Schneider, Klaus Pantel, Daniel J. Smit, Christoffer Gebhardt

**Affiliations:** 1Department of Dermatology and Venereology, University Medical Center Hamburg-Eppendorf, 20246 Hamburg, Germany; inka-lilott.hoehne@stud.uke.uni-hamburg.de (I.L.H.); i.heidrich@uke.de (I.H.); noah.zimmermann@stud.uke.uni-hamburg.de (N.Z.); tim.zell@stud.uke.uni-hamburg.de (T.Z.); g.geidel@uke.de (G.G.); a.ruenger@uke.de (A.R.); st.schneider@uke.de (S.W.S.); 2Fleur Hiege Center for Skin Cancer Research, University Medical Center Hamburg-Eppendorf, 20246 Hamburg, Germany; pantel@uke.de (K.P.); d.smit@uke.de (D.J.S.); 3Institute of Tumor Biology, University Medical Center Hamburg-Eppendorf, 20246 Hamburg, Germany; k.reese@uke.de

**Keywords:** melanoma, immunotherapy, biomarker, prognostic marker, overall survival, progression-free survival, liquid biopsy

## Abstract

**Simple Summary:**

In this prospective cohort study of metastatic melanoma patients receiving immune checkpoint inhibitors, we identified a serum chemokine CCL20 increase at baseline with a significantly impaired progression-free and overall survival and as an independent negative prognostic factor for PFS and OS in univariate as well as in multivariate analysis. CCL20 may represent a novel blood biomarker for the prediction of prognosis in advanced melanoma under immunotherapy with a special emphasis on progression.

**Abstract:**

Background: Immune checkpoint inhibition has revolutionized melanoma therapy, but many patients show primary or secondary resistance. Biomarkers are, therefore, urgently required to predict response prior to the initiation of therapy and to monitor disease progression. Methods: In this prospective study, we analyzed the serum C-C motif chemokine ligand 20 (CCL20) concentration using an enzyme-linked immunosorbent assay. Blood was obtained at baseline before the initiation of immunotherapy with anti-PD-1 monotherapy or Nivolumab and Ipilimumab in advanced melanoma patients (stages III and IV) enrolled at the University Medical Center Hamburg-Eppendorf. The CCL20 levels were correlated with clinico-pathological parameters and disease-related outcomes. Results: An increased C-C motif chemokine ligand 20 (CCL20) concentration (≥0.34 pg/mL) at baseline was associated with a significantly impaired progression-free survival (PFS) in the high-CCL20 group (3 months (95% CI: 2–6 months) vs. 11 months (95% CI: 6–26 months)) (*p* = 0.0033) and could be identified as an independent negative prognostic factor for PFS in univariate (Hazard Ratio (HR): 1.98, 95% CI 1.25–3.12, *p* = 0.004) and multivariate (HR: 1.99, 95% CI 1.21–3.29, *p* = 0.007) Cox regression analysis, which was associated with a higher risk than S100 (HR: 1.74). Moreover, high CCL20 levels were associated with impaired overall survival (median OS not reached for low-CCL20 group, *p* = 0.042) with an HR of 1.85 (95% CI 1.02–3.37, *p* = 0.043) in univariate analysis similar to the established prognostic marker S100 (HR: 1.99, 95% CI: 1.02–3.88, *p* = 0.043). Conclusions: CCL20 may represent a novel blood-based biomarker for the prediction of resistance to immunotherapy that can be used in combination with established strong clinical predictors (e.g., ECOG performance score) and laboratory markers (e.g., S100) in advanced melanoma patients. Future prospective randomized trials are needed to establish CCL20 as a liquid biopsy-based biomarker in advanced melanoma.

## 1. Introduction

As ultraviolet exposure is the most important risk factor for cutaneous melanoma, the incidence of cutaneous melanoma has risen sharply in recent decades, especially in predominantly fair-skinned populations [1]. In 2020, more than 325,000 new cases worldwide were recorded [2]. Due to the lack of early symptoms and the high metastasis rate, melanoma is a very aggressive tumor entity and is responsible for around 57,000 deaths per year worldwide [2]. Untreated patients with advanced melanoma achieve a 5-year survival rate of only 5–19% [3]. Since the establishment of targeted therapy and the introduction of immune checkpoint inhibition (ICI) in 2011, the treatment of melanoma has been revolutionized and the survival of patients has significantly improved. The standard therapy for patients with advanced melanoma is the combined treatment of Ipilimumab, a cytotoxic T-lymphocyte-associated protein 4 (CTLA-4) antibody, and Nivolumab, a programmed cell death protein 1 (PD-1) antibody, or a monotherapy with a PD-1 antibody (either Nivolumab or Pembrolizumab). These treatments reach a 5-year survival rate of 52% (Ipilimumab + Nivolumab) and 44% (PD-1 antibody monotherapy) [4]. Unfortunately, due to therapy resistance, not all melanoma patients benefit from their therapy, and in 50% of patients with advanced melanoma, the tumor progresses despite promising ICI therapy [5]. The Society for Immunotherapy of Cancer (SITC) distinguishes between primary therapy resistance by detecting tumor progression within the first six months of therapy, and secondary therapy resistance by developing tumor progression after six months of therapy [6]. These tumor progressions are detected using regular radiological staging, such as computer tomography (CT) scans, magnet resonance imaging (MRI) or positron emission tomography (PET). By the time the therapy resistance is identified through radiological staging, the prognosis of the patients is already impaired because of the rapid advancement of the tumor. Therefore, a method that could detect therapy resistance at an early time point is urgently needed to adapt the tumor therapy in order to prevent the further progression of the disease. An option for the identification of the high-risk cohort for progression could be liquid biopsy (LB) approaches [7]. By analyzing tumor components that are released into the blood by primary or metastatic cells, tumor characteristics including tumor entity or tumor development can be determined and monitored [8]. A blood-based biomarker that is minimally invasively accessible, determined repeatedly and reflects tumor characteristics in real time, would, therefore, be helpful in recognizing therapy resistance in an early stage.

Recently, the C-C motif chemokine ligand 20 (CCL20) and its specific chemokine receptor 6 (CCR6) have received tremendous attention in cancer research. CCL20, a small cytokine with an approximate molecular weight of 8 kDa with a total of 96 amino acids, also known as liver activation-regulated chemokine (LARC) and macrophage inflammatory protein-3 (MIP3A) is mainly secreted by immune cells (neutrophils, T lymphocytes, Th17 cells, B lymphocytes, natural killer cells, dendritic cells and macrophages) and is known to be involved in inflammatory processes [9,10]. Moreover, it has been reported that the CCL20/CCR6 signaling axis is critically involved in the pathogenesis of autoimmune diseases including rheumatoid arthritis, psoriasis and inflammatory bowel disease [9]. However, it was shown that CCL20/CCR6 signaling also plays an important role in immunomodulatory processes that can exert an oncogenic function. In various tumor entities, such as cervical carcinoma, pancreatic ductal adenocarcinoma, cutaneous squamous cell carcinoma and breast cancer, a high expression of CCL20 in the tumor tissue was associated with tumor progression [11,12,13,14]. In addition, it has been reported that among the immunosuppressive effect, CCL20 causes tumor progression by promoting crucial cellular processes including proliferation, invasion, angiogenesis and chemoresistance in several tumor entities [10,14,15]. Martin-Garcia et al. were also able to demonstrate the role of CCL20 and its receptor CCR6 in a mouse model by challenging with B16 melanoma cells [16].

In primary melanoma, Samaniego et al. reported in a cohort of 40 that high stromal levels of CCL20 predict poor survival [17]. Based on the observation in tissue, in this prospective study, we investigated whether the serum concentration of CCL20 in ICI-treated patients with advanced melanoma can predict the therapy response and overall survival (OS). 

## 2. Materials and Methods

### 2.1. Study Population

The study includes a total of 101 patients with advanced melanoma (cutaneous, mucosal and uveal) who received immunotherapy (Ipilimumab + Nivolumab, Nivolumab only, Pembrolizumab, Tebentafusp or Cemiplimab) at the Department of Dermatology and Venereology, Skin Cancer Center at the University Medical Center Hamburg-Eppendorf between 2018 and 2022. Detailed information on the cohort analyzed can be found in Table 1. Tumor stages were encoded according to the 8th edition of the American Joint Committee on Cancer (AJCC) melanoma staging system [18]. Performance stages were encoded according to the Eastern Cooperative Oncology Group (ECOG) Performance Status Scale [19]. In case intermediate scores (e.g., ECOG 1-2) have been reported, the higher stage was included in the analysis. Demographic, clinical and pathological data were retrieved from the clinical records at the University Medical Center Hamburg-Eppendorf. The study was approved by the Medical Ethical Committee, Hamburg, Germany and complies with the principles of the Declaration of Helsinki. Informed consent was obtained from all patients (PV5392). 

### 2.2. Collection of Peripheral Blood Samples

Blood samples of advanced melanoma patients were collected in serum containers (S-Monovette Serum Gel, Sarstedt, Germany) prior to the admission of therapy (referred to as baseline). Serum samples were centrifuged within 2 h after collection at 1800× *g* for 10 min and aliquoted before storage at −80 °C.

### 2.3. Enzyme-Linked Immunosorbent Assay (ELISA) for CCL20 Levels

The serum C-C motif chemokine ligand 20 (CCL20) levels were determined using the CCL20 (MIP-3α) Human ELISA Kit (#441404, BioLegend, San Diego, CA, USA). In brief, on the first day, the provided antibodies were coated according to the manufacturer’s instructions. The next day, unbound antibodies were washed from the microtiter plate. Thereafter, the samples were thawed on ice, diluted according to the manufacturer’s instructions and transferred to the microtiter plate. After incubation for 2 h at room temperature, the wells were washed, incubated with anti-CCL20 antibody and incubated again for 1 h at room temperature. Thereafter, the wells were washed, and the provided Avidin-HRP solution was added to the microtiter plate. The adsorption was measured at 450-nanometer and 570-nanometer wavelengths in a microplate reader (Power Wave XS2, BioTek, Winooski, VT, USA). Samples were analyzed in triplicates. A standard curve of the supplied recombinant CCL20 standard was created once per assay. Absorption values at 570 nm were subtracted prior to further analysis. CCL20 concentrations (pg/mL) of the patient samples were calculated according to the formula of the standard curve.

### 2.4. Statistical Analysis

The statistical analysis of the data was carried out using SPSS Statistics version 29 (IBM Inc., Armonk, NY, USA) and R version 4.3.2 (R Foundation for Statistical Computing, Vienna, Austria). The R packages used for analysis and visualization include ggplot2 version 3.3.4 [20], finalfit version 1.0.7 [21], survminer version 0.4.9 [22] and survival 3.5-7 [23,24]. Maximally selected rank statistics were calculated using the maxstat package version 0.7-25 [24,25]. 

Laboratory data below the lower limit of detection (i.e., S100, CRP and D-Dimers) were set to equal half of the lower limit of detection. Categorical variables were described using absolute numbers and percentages, and differences between groups were assessed using Fisher’s exact test. Continuous variables were tested for normality (Shapiro–Wilk test) and equality of variances (Levene test) where applicable. The means of two groups of unpaired samples with continuous variables were compared using Student’s *t*-test (parametric, equal variance), Welch’s *t*-test (parametric, unequal variance) or the Mann–Whitney U test (non-parametric) where applicable. For OS and progression-free survival (PFS) analysis, the Kaplan–Meier method was used, and the statistical analysis was conducted using the log-rank test (Mantel–Cox). To determine the predictors of PFS and OS, the Cox Proportional Hazards Regression (univariate and multivariate) was used. For the multivariate analysis of PFS and OS, parameters that were significant in univariate analysis were included (i.e., ECOG score, CCL20 group and S100). A *p*-value of < 0.05 was considered statistically significant.

## 3. Results

As the analysis of serum C-C motif chemokine ligand 20 (CCL20) is not standard in clinical practice, and, therefore, no reference range exists, we determined the median CCL20 levels of the 101 advanced melanoma patients at baseline prior to the initiation of immunotherapy. The CCL20 levels ranged from 0 pg/mL to 35.19 pg/mL with a median of 0.26 pg/mL (IQR: 3.1 pg/mL) in our advanced melanoma study cohort. In order to find the optimal cutoff for prognosis based on the measured CCL20 concentrations, we analyzed the optimal cutoff through maximally selected rank statistics using the maxstat package with progression-free survival (PFS) time as an input. The optimal cutoff calculated was 0.34 pg/mL. Stratification according to the suggested cutoff resulted in 54 melanoma patients in the low-CCL20 group (<0.34 pg/mL) and 47 in the high-CCL20 group (≥0.34 pg/mL). We additionally evaluated the median split as well as the 25% and 75% quantiles for cutoff determination; however, maximally selected rank statistics resulted in better risk stratification and were, therefore, used. The analysis using the log-rank test demonstrated that melanoma patients with high CCL20 have an impaired PFS with a median of 3 months (95% CI: 2–6) compared to patients with a low CCL20 with a median of 10.5 months (95% CI: 6–26) (*p* = 0.0033) (Figure 1). 

With respect to OS, we observed that melanoma patients with high CCL20 levels have shorter survival times compared to patients with low CCL20 levels (median OS not reached in the low-CCL20 group) (*p* = 0.042) (Figure 2).

We then further analyzed the CCL20 subgroups with respect to demographic and clinico-pathological parameters. No differences were observed for the sex (*p* = 0.521) and Eastern Cooperative Oncology Group (ECOG) Performance Status Scale (*p* = 0.241) of the patients included according to CCL20 group, whereas the age groups (<65 and ≥65) were significantly different between the low- and high-CCL20 groups (*p* = 0.024). Moreover, no differences regarding the primary tumor characteristics were observed regarding primary the melanoma site (*p* = 0.060), AJCC stage (*p* = 0.721), tumor size (T) (*p* = 0.782), lymph node positivity (N) (*p* = 0.600) and presence of distant metastasis (M) before therapy (*p* = 0.537). Baseline therapy was not significantly altered in the low- and high-CCL20 group (*p* = 0.686). CCL20 levels increased from ECOG 0 to ECOG 3 (Appendix A) and were higher in AJCC stage IV than in AJCC stage III patients (Appendix A). The number of patients with progressive disease was significantly higher in the high-CCL20 group (*p* = 0.024) (Table 2).

With respect to laboratory characteristics, we observed a significantly higher LDH as well as S100 in the high-CCL20 group (*p* = 0.004 and *p* = 0.049, respectively). Regarding D-Dimers (*p* = 0.489), CRP (*p* = 0.076), neutrophil count (*p* = 0.278), lymphocyte count (*p* = 0.192) and the neutrophil/lymphocyte ratio (NLR) (*p* = 0.167), no significant differences were observed between the groups (Table 3).

Cox regression was performed in order to determine the independent prognosticators of progression-free survival (Table 4). In univariable regression, sex, age group, AJCC stage, primary melanoma site, baseline therapy and elevated LDH levels at baseline did not significantly impact PFS, whereas ECOG, CCL20 group and elevated S100 at baseline increased the risk of an impaired PFS. With respect to ECOG scores at baseline, an increasing risk could be detected for ECOG 1 (HR 1.34 95% CI 0.81–2.24) (*p* = 0.257), ECOG 2 (HR 1.96 95% CI 0.97–3.98) (*p* = 0.062) and ECOG 3 (HR 7.03 95% CI 2.63–18.80) (*p* < 0.001) compared to ECOG 0 patients. However, only ECOG 3 was significant. Elevated S100 levels at baseline are associated with an HR of 1.74 (95% CI 1.05–2.86) (*p* = 0.030). Patients in the high-CCL20 group had an HR of 1.98 (95% CI 1.25–3.12) (*p* = 0.004) regarding progression compared to the low-CCL20 group (Table 4). 

All the significant variables from the univariable Cox regression persisted in multivariate Cox regression (ECOG score 3, elevated S100 and high-CCL20 group). ECOG score, in particular ECOG 3, had a significantly higher HR of 9.48 (95% CI 3.36–26.75) (*p* < 0.001). The high-CCL20 group contributed to a moderate risk increase with an HR of 1.99 (95% CI 1.21–3.29) (*p* = 0.007) regarding PFS in the advanced melanoma cohort, which is a stronger risk factor than elevated S100 at baseline (HR: 1.74, 95% CI: 1.05–2.90, *p* = 0.033) (Table 4). 

Next, we analyzed the impact of CCL20 on overall survival. Univariable Cox regression revealed that ECOG > 0 (ECOG 1 HR 4.20, 95% CI 1.95–9.07, *p* < 0.001; ECOG 2 HR 7.87, 95% CI 3.24–19.11, *p* < 0.001; ECOG 3 HR 16.14, 95% CI 5.26–49.53, *p* < 0.001), the high-CCL20 group (HR 1.85, 95% CI 1.02–3.37, *p* = 0.043) and an elevated S100 at baseline (HR 1.99, 95% CI 1.02–3.88, *p* = 0.043) are risk factors for decreased OS in the advanced melanoma cohort. Age group, AJCC stage, primary melanoma site, baseline therapy and LDH at baseline did not show a statistically significant effect in univariate Cox proportional hazard analysis. In the next step, multivariate analysis was conducted. Regarding OS only, ECOG > 0 (ECOG 1 HR 4.58, 95% CI 1.97–10.60, *p* < 0.001; ECOG 2 HR 7.46, 95% CI 2.75–20.24, *p* < 0.001; ECOG 3 HR 15.36, 95% CI 4.75–49.67, *p* < 0.001) persisted in multivariate analysis. In the multivariate Cox regression proportional hazard model, a strong trend could be observed for elevated S100 at baseline (HR 1.78, 95% CI 0.90–3.54, *p* = 0.098) but not for the high-CCL20 group (HR 1.46, 95% CI 0.77–2.75, *p* = 0.244) regarding OS (Table 5).

Moreover, we evaluated whether the combination of CCL20 and already-established prognostic markers (i.e., S100) could contribute to improved risk stratification. With respect to PFS, we observed that patients with low CCL20 and low S100 showed the longest time to progression with a median of 19 months (95% CI: 10-infinite) compared to the patients with high CCL20 and elevated S100 with a median of only 2 months (95% CI: 1–6). Having either high CCL20 and low S100 or low CCL20 and high S100 resulted in an intermediate PFS time (both were 7.5 months, 95% CI: 2-infinite) (*p* = 0.0075) (Figure 3A). Regarding OS, a similar trend could be observed with a lower median OS in the high-CCL20 /high-S100 group with a median of 10.5 months (95% CI: 5-infinite) compared to the low-CCL20 /low-S100 group (median OS not reached) (Figure 3B). 

## 4. Discussion

Even though the treatment of melanoma patients has tremendously improved over the last number of years, approximately half of the patients with advanced melanoma do not benefit from ICI therapy due to therapy resistance [4,5]. A blood-based biomarker that detects tumor progression in an early state is, therefore, urgently required. CCL20 has become of great interest in tumor research as it promotes tumor progression in different solid tumor entities, including cervical carcinoma, pancreatic ductal adenocarcinoma, cutaneous squamous cell carcinoma and breast cancer [11,12,13,14]. The presence of CCL20 has been associated with an immunosuppressive environment, thereby attenuating the effect of ICI therapy. Moreover, it has been reported that CCL20 can stimulate the proliferation, invasion, angiogenesis and therapy resistance of cancer cells, thereby facilitating tumor growth [10,14,15]. In a study by Wang et al., the authors discovered that the serum CCL20 concentration can be used as an early detection and prognostic biomarker in colorectal carcinoma [26]. In line with this, it was shown in the mouse model that B16 melanoma cells with CCL20 in CCR-sufficient mice lead to larger tumors compared to the injection of B16 melanoma cells without CCL20. Furthermore, tumor growth was most inefficient in CCR6-/- knockout mice [16]. As the role of CCL20 in humans has only been demonstrated in tissue samples for melanoma in the past [17], we investigated whether CCL20 could also serve as a blood-based prognostic biomarker in melanoma patients. In line with the previous report on CCL20, we observed that high serum CCL20 concentrations before therapy are associated with significantly impaired PFS (*p* = 0.0033) and significantly lower overall survival rates (*p* = 0.042) in this single-center advanced melanoma cohort. In this work, we demonstrate that the CCL20 concentration at baseline (before therapy) represents an independent, prognostic factor for PFS, analyzed in the univariate (HR: 1.98, 95% CI: 1.25–3.12, *p* = 0.004) as well as in the multivariate analysis (HR: 1.99, 95% CI: 1.21–3.29, *p* = 0.007). Consequently, a high serum CCL20 concentration before therapy is an independent risk factor for tumor progression. 

Furthermore, the CCL20 serum concentration before therapy represents an independent, prognostic factor for OS; however, this is only in the univariate analysis (HR: 1.85, 95% CI: 1.02–3.37, *p* = 0.043). Due to the high relevance of ECOG considering OS, the CCL20 serum concentration plays a subordinate role. Nevertheless, high CCL20 serum concentrations are significantly associated with a shorter OS in this cohort. 

In addition, our analysis has shown that LDH and S100, which are already implemented as laboratory characteristics to monitor melanoma patients, are significantly elevated in the high-CCL20 patient group (*p* = 0.004 and *p* = 0.049, respectively). We also demonstrated that the combination of these two blood-based markers (i.e., CCL20 and S100) leads to improved risk stratification with a strong decrease in PFS and OS time in particular in the high-CCL20/high-S100 subgroup. 

Despite the demonstrated prognostic role of CCL20 in previous work, as well as our study, it would also be important to decipher the molecular mechanism in melanoma patients. Due to the observational nature of our study, the results do not allow for the interpretation of the causality and the molecular mechanism of CCL20 in melanoma patients. Moreover, in future work, the dynamics of CCL20 should be investigated to further understand the role of this cytokine for melanoma patients undergoing ICI. Understanding the underlying mechanisms could support the identification of novel targets for melanoma therapy and might be a suitable approach for overcoming ICI-related therapy resistance. In the past, it has been demonstrated that melanoma cell lines express the CCL20 receptor C-C chemokine receptor type 6 (CCR6) and that the tumor-associated macrophages (TAMs) are the main source of CCL20 expression. Subsequently, CCL20 binding to melanoma cells would then be responsible for the rapid progression of the disease [17]. The fact that TAMs are the main source of measured CCL20 in the serum of these patients could explain why elderly patients (≥65 years) have significantly lower CCL20 concentrations compared to younger patients (<65 years) (*p* = 0.024) in our present study. With respect to this, it has been demonstrated that the immune system ages during the course of life and is less active in older people [27]. Our finding of CCL20 being an independent prognostic marker can be used as a biomarker for melanoma patients and has clinical relevance, as statements about the prognosis and therapy response can be made based on this. This provides a basis for further therapeutic actions and improves individualized patient care. With different treatment options, the knowledge of ICI response can support clinical decision making to find the best possible treatment for the patient and counteract tumor progression at an early stage. Moreover, approximately 82% to 95% of ICI-treated patients develop side effects, whereby one-third have to interrupt or terminate the treatment due to serious immunotherapy-related adverse effects [28].

The regulation of CCL20 secretion has not been fully elucidated yet. Interestingly, a common expression with, e.g., growth hormones including epidermal growth factor (EGF) has been reported, which has been shown to play a crucial role in melanoma pathogenesis [29,30]. The strong association with other oncogenic stimuli could enhance tumor growth in high-CCL20 tumors and patients. In hepatocellular carcinoma (HCC), a study by Hou et al. reported that CCL20 induces epithelial–mesenchymal transition (EMT) in HCC cell lines and activates the PI3K/AKT/mTOR pathway and the Wnt/β-catenin signaling pathway, thereby promoting proliferation and migration [15]. Moreover, in a study by Fenouille et al., it has been reported that the PI3K/AKT/mTOR signaling pathway induces EMT in melanocytes [31]. Another study by Madhunapantula et al. additionally reports that the PI3K/AKT pathway inhibits the cell senescence and apoptosis of melanocytes and thereby promotes melanogenesis [32]. These studies demonstrate that the oncogenic PI3K/AKT/mTOR pathway plays an important role in the progression of melanoma. As this signaling pathway is activated by CCL20 in hepatocellular carcinoma (HCC), it could be assumed that this occurs in melanoma cells as well, particularly CCL20 signals via the PI3K/AKT/mTOR pathway. Furthermore, in uveal melanoma, it has also been demonstrated that the activation of the Wnt/β-catenin signaling pathway contributes to an immunosuppressive environment via the recruitment of regulatory T cells and directly stimulates tumor cells to proliferate, differentiate and metastasize [33,34,35,36]. With regard to the reported activation of the Wnt/β-catenin signaling pathway via CCL20 in HCC, this may be a potential explanation for the impaired prognosis of high-CCL20 melanoma patients. 

The understanding of the mechanisms behind the actions of CCL20 is important to develop new therapies for tumor patients. With respect to CCL20 inhibition, it has already been reported that CCL20 inhibition improves outcomes significantly in cutaneous squamous cell carcinoma patients receiving radiotherapy [37]. 

## 5. Conclusions

In conclusion, CCL20 is a promising blood-based biomarker for therapeutic response and ICI in advanced melanoma. The combination with the established melanoma marker S100 can further improve risk stratification. Further functional studies are required to gain a greater understanding of CCL20-associated signaling, demonstrate clinical impact in larger (multi-center) cohorts and discover potential new therapeutic targets, especially for advanced melanoma with resistance to ICI.

## Figures and Tables

**Figure 1 cancers-16-01737-f001:**
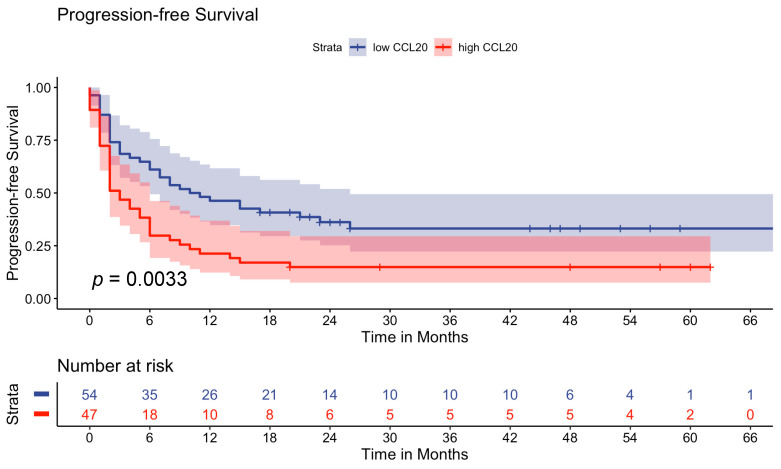
Kaplan–Meier survival curves displaying time to progression in advanced melanoma patients (n = 101). Univariate analysis was carried out via the log-rank test (Mantel–Cox). A *p*-value < 0.05 was considered statistically significant.

**Figure 2 cancers-16-01737-f002:**
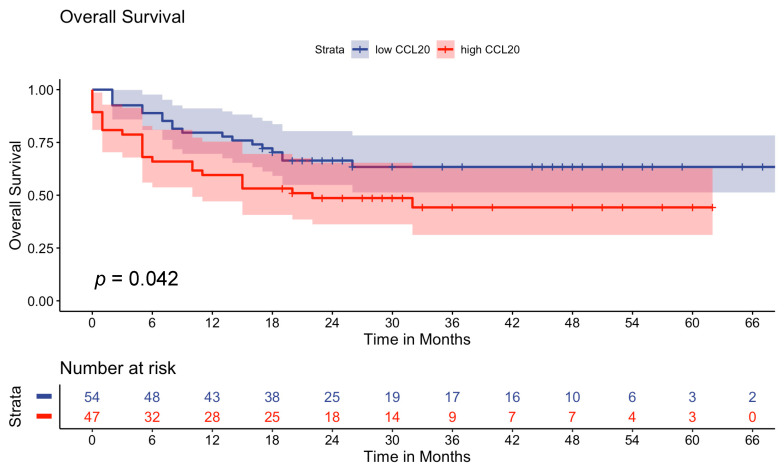
Kaplan–Meier survival curves displaying the overall survival in advanced melanoma patients (n = 101). Univariate analysis was carried out via the log-rank test (Mantel–Cox). A *p*-value < 0.05 was considered statistically significant.

**Figure 3 cancers-16-01737-f003:**
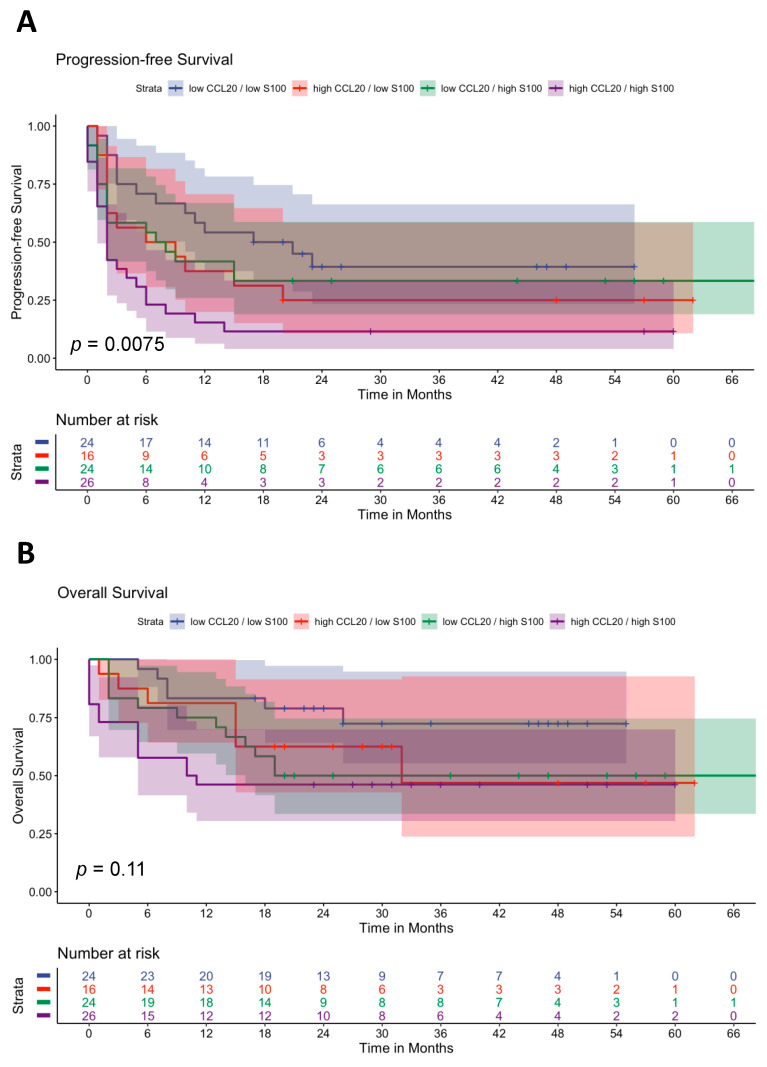
Kaplan–Meier survival curves displaying the progression-free survival (**A**) and overall survival (**B**) in advanced melanoma patients after subgroup analysis of CCL20 and S100. In total, 90 (out of 101) patients had S100 measured at baseline and were included in this analysis. Univariate analysis was carried out via the log-rank test (Mantel–Cox). A *p*-value < 0.05 was considered statistically significant.

**Table 1 cancers-16-01737-t001:** Overview of patient demographics and clinico-pathological parameters of 101 melanoma patients at baseline. ECOG: Eastern Cooperative Oncology Group; AJCC: American Joint Committee on Cancer; T: tumor size, N: lymph node positivity, M: distant metastasis (according to the TNM classification).

	Total N (%)		N (%)
Sex	101 (100.0)	Male	69 (68.3)
		Female	32 (31.7)
Age group	101 (100.0)	<65	39 (38.6)
		≥65	62 (61.4)
ECOG	101 (100.0)	0	51 (50.5)
		1	34 (33.7)
		2	11 (10.9)
		3	5 (5.0)
Primary melanoma site	101 (100.0)	Cutaneous	73 (72.3)
		Mucosal	8 (7.9)
		Uveal	20 (19.8)
AJCC Stage	101 (100.0)	Stage III	8 (7.9)
		Stage IV	93 (92.1)
T	101 (100.0)	T0	19 (18.8)
		T1	7 (6.9)
		T2	11 (10.9)
		T3	11 (10.9)
		T4	33 (32.7)
		Tx	20 (19.8)
N	101 (100.0)	N0	41 (40.6)
		N1	23 (22.8)
		N2	14 (13.9)
		N3	18 (17.8)
		Nx	5 (5.0)
M	101 (100.0)	M0	11 (10.9)
		M1	90 (89.1)
Baseline therapy	101 (100.0)	Ipilimumab + Nivolumab	64 (63.4)
		Nivolumab	12 (11.9)
		Pembrolizumab	16 (15.8)
		Tebentafusp	8 (7.9)
		Cemiplimab	1 (1.0)

**Table 2 cancers-16-01737-t002:** Baseline clinical characteristics of the advanced melanoma cohort according to CCL20 group. ECOG: Eastern Cooperative Oncology Group; AJCC: American Joint Committee on Cancer; T: tumor size; N: lymph node positivity; M: distant metastasis (according to the TNM classification).

	Total N (%)		Low CCL20 < 0.34 pg/mL N (%)	High CCL20 ≥ 0.34 pg/mL N (%)	*p*-Value (Fisher’s Exact Test)
Sex	101 (100.0)	Male	35 (64.8)	34 (72.3)	0.521
		Female	19 (35.2)	13 (27.7)	
Age group	101 (100.0)	<65	15 (27.8)	24 (51.1)	**0.024**
		≥65	39 (72.2)	23 (48.9)	
ECOG	101 (100.0)	0	32 (59.3)	19 (40.4)	0.241
		1	14 (25.9)	20 (42.6)	
		2	5 (9.3)	6 (12.8)	
		3	3 (5.6)	2 (4.3)	
Primary melanoma site	101 (100.0)	Cutaneous	43 (79.6)	30 (63.8)	0.060
		Mucosal	5 (9.3)	3 (6.4)	
		Uveal	6 (11.1)	14 (29.8)	
AJCC Stage	101 (100.0)	Stage III	5 (9.3)	3 (6.4)	0.721
		Stage IV	49 (90.7)	44 (93.6)	
T	101 (100.0)	T0	9 (16.7)	10 (21.3)	0.782
		T1	4 (7.4)	3 (6.4)	
		T2	8 (14.8)	3 (6.4)	
		T3	6 (11.1)	5 (10.6)	
		T4	18 (33.3)	15 (31.9)	
		Tx	9 (16.7)	11 (23.4)	
N	101 (100.0)	N0	19 (35.2)	22 (46.8)	0.600
		N1	12 (22.2)	11 (23.4)	
		N2	9 (16.7)	5 (10.6)	
		N3	10 (18.5)	8 (17.0)	
		Nx	4 (7.4)	1 (2.1)	
M	101 (100.0)	M0	7 (13.0)	4 (8.5)	0.537
		M1	47 (87.0)	43 (91.5)	
Baseline Therapy	101 (100.0)	Ipilimumab + Nivolumab	35 (64.8)	29 (61.7)	0.686
		Nivolumab	6 (11.1)	6 (12.8)	
		Pembrolizumab	10 (18.5)	6 (12.8)	
		Tebentafusp	3 (5.6)	5 (10.6)	
		Cemiplimab	0 (0.0)	1 (2.1)	
Progressive Disease	101 (100.0)	No	19 (35.2)	7 (14.9)	**0.024**
		Yes	35 (64.8)	40 (85.1)	

**Table 3 cancers-16-01737-t003:** Baseline laboratory characteristics of the advanced melanoma cohort according to CCL20 group. (a) Mann–Whitney U test, (b) Student’s *t*-test. LDH: lactate dehydrogenase; S100; CRP: C-reactive protein; NLR: Neutrophil/Lymphocyte ratio.

Total N (%)		Low CCL20 < 0.34 pg/mL	High CCL20 ≥ 0.34 pg/mL		*p*-Value
LDH [U/L]	98 (97.0)	Median (IQR)	258 (220.75 to 322.50)	329 (262.25 to 575.00)	**0.004 (a)**
S100 [µg/L]	90 (89.1)	Median (IQR)	0.147 (0.06 to 0.48)	0.257 (0.09 to 1.16)	**0.049 (a)**
D-Dimers [mg/L]	83 (82.2)	Median (IQR)	0.915 (0.58 to 1.73)	1.170 (0.47 to 3.30)	0.489 (a)
CRP [mg/L]	96 (95.0)	Median (IQR)	5 (2.50 to 25.00)	10 (5.00 to 34.00)	0.076 (a)
Neutrophil count [×10^9^/L]	93 (92.1)	Median (IQR)	4.830 (3.92 to 5.88)	5.275 (4.28 to 7.46)	0.278 (a)
Lymphocyte count [×10^9^/L]	93 (92.1)	Mean (SD)	1.486 (0.61)	1.330 (0.54)	0.192 (b)
Neutrophil/Lymphocyte ratio (NLR)	93 (92.1)	Median (IQR)	3.199 (2.65 to 5.31)	3.854 (2.86 to 6.69)	0.167 (a)

**Table 4 cancers-16-01737-t004:** Univariate and multivariate Cox proportional hazard analysis with respect to progression-free survival in advanced melanoma patients (n = 101). ECOG: Eastern Cooperative Oncology Group; AJCC: American Joint Committee on Cancer; T: tumor size; N: lymph node positivity; M: distant metastasis (according to the TNM classification).

Progression-Free Survival		N (%)	HR (Univariable) (95% CI)	HR (Multivariable) (95% CI)
Sex	Male	69 (68.3)	-	-
	Female	32 (31.7)	0.67 (0.40–1.12, *p* = 0.129)	-
Age group	<65	39 (38.6)	-	-
	≥65	62 (61.4)	0.86 (0.54–1.36, *p* = 0.508)	-
ECOG	0	51 (50.5)	-	-
	1	34 (33.7)	1.34 (0.81–2.24, *p* = 0.257)	1.54 (0.89–2.67, *p* = 0.127)
	2	11 (10.9)	1.96 (0.97–3.98, *p* = 0.062)	2.18 (0.98–4.83, *p* = 0.056)
	3	5 (5.0)	**7.03 (2.63–18.80** **, *p* < 0.001)**	**9.48** **(3.36–26.75, *p* < 0.001)**
AJCC Stage	Stage III	8 (7.9)	-	-
	Stage IV	93 (92.1)	1.70 (0.69–4.22, *p* = 0.252)	-
Primary melanoma site	Cutaneous	73 (72.3)	-	-
	Mucosal	8 (7.9)	1.80 (0.81–4.00, *p* = 0.151)	-
	Uveal	20 (19.8)	1.57 (0.90–2.72, *p* = 0.111)	-
Baseline therapy	Ipilimumab + Nivolumab	64 (63.4)	-	-
	Nivolumab	12 (11.9)	1.12 (0.56–2.22, *p* = 0.752)	-
	Pembrolizumab	16 (15.8)	0.91 (0.48–1.71, *p* = 0.760)	-
	Tebentafusp	8 (7.9)	1.37 (0.59–3.22, *p* = 0.466)	-
	Cemiplimab	1 (1.0)	7.24 (0.95–55.01, *p* = 0.056)	-
CCL20 group	Low-CCL20 < 0.34 pg/mL	54 (53.5)	-	-
	High-CCL20 ≥ 0.34 pg/mL	47 (46.5)	**1.98 (1.25–3.12, *p* = 0.004)**	**1.99 (1.21–3.29, *p* = 0.007)**
LDH	Not elevated	28 (28.6)	-	-
	Elevated	70 (71.4)	0.91 (0.55–1.51, *p* = 0.720)	-
S100	Not elevated	40 (44.4)	-	-
	Elevated	50 (55.6)	**1.74 (1.05–2.86, *p* = 0.030)**	**1.74 (1.05–2.90, *p* = 0.033)**

**Table 5 cancers-16-01737-t005:** Univariate and multivariate Cox proportional hazard analysis with respect to overall survival in advanced melanoma patients (n = 101). ECOG: Eastern Cooperative Oncology Group; AJCC: American Joint Committee on Cancer; T: tumor size, N: lymph node positivity, M: distant metastasis (according to the TNM classification).

Overall Survival		N (%)	HR (Univariable) (95% CI)	HR (Multivariable) (95% CI)
Sex	Male	69 (68.3)	-	-
	Female	32 (31.7)	0.71 (0.36–1.40, *p* = 0.317)	-
Age group	<65	39 (38.6)	-	-
	≥65	62 (61.4)	1.08 (0.59–1.98, *p* = 0.811)	-
ECOG	0	51 (50.5)	-	-
	1	34 (33.7)	**4.20 (1.95–9.07, *p* < 0.001)**	**4.58** **(1.97–10.60, *p* < 0.001)**
	2	11 (10.9)	**7.87 (3.24–19.11, *p* < 0.001)**	**7.46** **(2.75–20.24, *p* < 0.001)**
	3	5 (5.0)	**16.14** **(5.26–49.53, *p* < 0.001)**	**15.36** **(4.75–49.67, *p* < 0.001)**
AJCC	Stage III	8 (7.9)	-	-
	Stage IV	93 (92.1)	2.03 (0.49–8.40, *p* = 0.327)	-
Primary melanoma site	Cutaneous	73 (72.3)	-	-
	Mucosal	8 (7.9)	1.77 (0.68–4.58, *p* = 0.241)	-
	Uveal	20 (19.8)	1.48 (0.72–3.06, *p* = 0.283)	-
Baseline therapy	Ipilimumab + Nivolumab	64 (63.4)	-	-
	Nivolumab	12 (11.9)	0.72 (0.25–2.05, *p* = 0.537)	-
	Pembrolizumab	16 (15.8)	1.08 (0.50–2.37, *p* = 0.838)	-
	Tebentafusp	8 (7.9)	0.57 (0.14–2.40, *p* = 0.444)	-
	Cemiplimab	1 (1.0)	5.06 (0.67–38.26, *p* = 0.117)	-
CCL20 group	Low-CCL20 < 0.34 pg/mL	54 (53.5)	-	-
	High-CCL20 ≥ 0.34 pg/mL	47 (46.5)	**1.85 (1.02–3.37, *p* = 0.043)**	1.46 (0.77–2.75, *p* = 0.244)
LDH	Not elevated	28 (28.6)	-	-
	Elevated	70 (71.4)	1.16 (0.58–2.31, *p* = 0.680)	-
S100	Not elevated	40 (44.4)	-	-
	Elevated	50 (55.6)	**1.99 (1.02–3.88, *p* = 0.043)**	1.78 (0.90–3.54, *p* = 0.098)

## Data Availability

The raw data supporting the conclusions of this article will be made available by the authors upon request.

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
