# Peer review of "High Serum Levels of CCL20 Are Associated with Recurrence and Unfavorable Overall Survival in Advanced Melanoma Patients Receiving Immunotherapy"

_cancers, 2024, doi:10.3390/cancers16091737_

Round 1

Reviewer 1 Report

Comments and Suggestions for Authors

The authors have tried to show the potential use of CCL20 as a biomarker for ICI response in melanoma. It is a good study and has been carried out properly, but the interpretation and /or presentation should be modified to improve the paper. I have a few suggestions:

1.       The abstract is written in a way that will tell the readers that baseline serum CCL20 is “the” predictor without mentioning anything about the ECOG. But in fact, ECOG (that needs only clinical assessment) has several folds stronger effect size both for PFS and OS even after adjustment for CCL20. I appreciate that the authors have mentioned this in the “Discussion” section. However, in the current format, the abstract may be misleading.

2.        Better to show the baseline CCL20 by ECOG status in a table. Also, it would be better to mention / show the follow-up CCL20 after starting the ICI therapy if available.

3.       Speaking about clinically usable biomarker, it is important to investigate and report the PPV, NPV in this case or even Sensitivity and Specificity.

4.       It is good that the authors have discussed the probable role of CCL20 in melanoma, but that is perhaps more relevant if they had CCL20 in the tissue level. Serum level of CCL20 may have less impact on pathogenicity, rather it may reflect the effect or immune response.

5.       It looks like, for the multivariate Cox regression model for PFS, the authors entered age group, ECOG, primary melanoma site, and CCL20 in the model. Better to mention that in the statistical method section. Similarly, for Cox regression for OS, they used age group, ECOG, CCL20 and S100.

Author Response

Dear reviewer 1,

we would like to thank you for the positive comments on our manuscript. Please find below our response to your suggestions. We hope that we could address your concerns to your satisfaction in this revised version.

Reviewer 2 Report

Comments and Suggestions for Authors

The aim of this prospective cohort study was to assess how the serum concentration of CCL20 in ICI treated patients with advanced melanoma can predict the therapy response and overall survival ratio. The work is of very high quality, both in terms of scientific novelty and editorial side. In my opinion this article should be published after revisions, listed below.

My major question is why the Authors haven’t checked how the choice of therapy, either combined treatment of Ipilimumab and Nivolumab, or a monotherapy with a PD-1 antibody affects the changes in the CCL20 levels?

I really appreciate the graphical abstract-it is informative and simply beautiful. Well done! 😊

Line 66, I suggest adding abbreviations (CT, MRI, PET) as they are well established acronyms

Line 77, here, it should be stated that CCL20 is also known as liver activation regulated chemokine (LARC) or Macrophage Inflammatory Protein-3 (MIP3A)

Line 77, in this paragraph a short sentence about the structure of CCL20 (i.e. number of residues) would be beneficial, also please consider adding the figure with the structure, i.e. from PDB

Line 99-101, the data on patients is quite limited, it should be extended and presented, preferably, in a form of a table

Line 159 – this must be an error and should be removed

Table 1 and Table 2 captions, the symbols (i.e. LDH, S100) must be explained

Line 240, this statement requires a reference

In the discussion section the Authors should add a sentence or two about the limitations of the current study

Author Response

Dear reviewer 2,   thank you very much for your kind words and constructive feedback on our manuscript. Please find below our point-by-point response to the questions you raised. We hope that we were able to answer them to your satisfaction.

Reviewer 3 Report

Comments and Suggestions for Authors

1. Line 159 - part of the text is missing, please correct the typo.

2. The blood level of CCL20 was 0.26 pg/mL ± 6.399 standard deviation (SD). I have many questions about this entry: with the range of CCL20 varying from 0 to 35.19 pg/mL, the median is very low. What is the reason for this dispersion? The authors do not analyze the original data for the study group, but immediately proceed to the median. Why calculate the standard deviation if the distribution is definitely not normal?

3. The authors use the CCL20 median as a cutoff point, but given the range of values, it may be appropriate to try, for example, the 25% or 75% quartile as a cutoff point. There is no detailed analysis that changing the cutoff point will not produce better data than described in the article. Justification for this threshold value is needed.

4. In Table 2, the authors also provide the mean and standard deviation, although for CCL20 they always mention the median, is this really the median? or is it an average everywhere? Why are different methods of statistical processing used: parametric and non-parametric together? Provide a check of the nature of the distribution.

5. In the group with high CCL20, the S100 level increases significantly; why is it not used to calculate the treatment prognosis? Or you can use a combination of these two markers. Similar question regarding LDH. By the way, the need to determine other blood parameters is not substantiated in any way in the article.

6. Based on the data in tables 3 and 4, I would suggest the authors to select a group according to 2 criteria: high CCL20 + high S100 and see how the forecast changes compared to the group in which both indicators are low.

In general, the article is interesting, but I have many questions about statistical processing.

Author Response

Dear reviewer 3,

thank you very much for the critical evaluation of our manuscript and your valuable suggestions. Please find our reply to your suggestion below.

Round 2

Reviewer 2 Report

Comments and Suggestions for Authors

The Authors have improved their work and current version should be accepted.

Author Response

Many thanks again to reviewer 2.

Reviewer 3 Report

Comments and Suggestions for Authors

The authors took my comments into account and significantly revised the manuscript in terms of improving statistical data processing.

Small notes:

1. In Table 3, indicate the units of measurement in the first column

2. Line 171 - in a little more detail you need to indicate how the optimal value of the cut-off point was determined.

Author Response

Many thanks to review 3 again.

Table 3:

We ad the units of measurement in the first column in table 3.

Line 171: 

In order to explain in more detail how the optimum value of the cut-off point was determined, we have adapted the text accordingly.

In order to find the optimal cut off for prognosis based on the measured CCL20 concentrations, we analyzed the optimal cut off by maximally selected rank statistics using the maxstat package with progression-free survival (PFS) time as an input. The optimal cut off calculated was 0.34 pg/mL. Stratification according to the suggested cut off resulted in 54 melanoma patients in the low CCL20 group (< 0.34 pg/mL) and 47 in the high CCL20 group (≥ 0.34 pg/mL). We additionally evaluated the median split as well as the 25% and 75% quantile for cut off determination, however, maximally selected rank statistics resulted in a better risk stratification and was therefore used.